# Advanced Techniques for the Intelligent Diagnosis of Fish Diseases: A Review

**DOI:** 10.3390/ani12212938

**Published:** 2022-10-26

**Authors:** Daoliang Li, Xin Li, Qi Wang, Yinfeng Hao

**Affiliations:** 1National Innovation Center for Digital Fishery, China Agricultural University, 17 Tsinghua East Road, Beijing 100083, China; 2College of Information and Electrical Engineering, China Agricultural University, Beijing 100083, China; 3Beijing Engineering and Technology Research Centre for Internet of Things in Agriculture, China Agriculture University, Beijing 100083, China; 4Key Laboratory of Agricultural Information Acquisition Technology, Ministry of Agriculture, Beijing 100083, China

**Keywords:** computer vision, fish disease, image processing, intelligent diagnosis, real-time detection

## Abstract

**Simple Summary:**

The automatic diagnosis of fish diseases is an important task in smart aquaculture, and plays a key role in detecting fish conditions, understanding disease signs, and improving fish welfare and health. This paper reviews the latest advances in intelligent fish disease diagnosis techniques for aquatic applications. To acquire high-quality images to improve the accuracy of fish disease diagnosis, the paper reviews, in the second paragraph, how image-processing techniques can be used to acquire high-quality images. The third paragraph of this paper reviews the methods of intelligent fish disease diagnosis, including expert system detection, camera image detection, microscope image detection, spectral image detection, fluorescence image detection, ultrasonic image detection, and sensor detection methods. Additionally, the advantages and disadvantages of each detection method are summarized, including whether they cause damage to the fish body. Finally, we put forward the prospect of the intelligent detection of fish diseases and summarize the methods that may be used in the intelligent diagnosis of fish disease.

**Abstract:**

Aquatic products, as essential sources of protein, have attracted considerable concern by producers and consumers. Precise fish disease prevention and treatment may provide not only healthy fish protein but also ecological and economic benefits. However, unlike intelligent two-dimensional diagnoses of plants and crops, one of the most serious challenges confronted in intelligent aquaculture diagnosis is its three-dimensional space. Expert systems have been applied to diagnose fish diseases in recent decades, allowing for restricted diagnosis of certain aquaculture. However, this method needs aquaculture professionals and specialists. In addition, diagnosis speed and efficiency are limited. Therefore, developing a new quick, automatic, and real-time diagnosis approach is very critical. The integration of image-processing and computer vision technology intelligently allows the diagnosis of fish diseases. This study comprehensively reviews image-processing technology and image-based fish disease detection methods, and analyzes the benefits and drawbacks of each diagnostic approach in different environments. Although it is widely acknowledged that there are many approaches for disease diagnosis and pathogen identification, some improvements in detection accuracy and speed are still needed. Constructing AR 3D images of fish diseases, standard and shared datasets, deep learning, and data fusion techniques will be helpful in improving the accuracy and speed of fish disease diagnosis.

## 1. Introduction

In recent years, aquatic products have played an essential role in the global food supply chain [1], which has a significant impact on the economy and the social development of developing countries. Fish, as an important source of protein, have a balanced nutrition ratio of essential elements required for the human body [2], continuously increasing the consumption of fish products [3]. With the increase in density and intensive development of aquatic fish farming, fish health has become a top issue of concern for consumers [4]. In addition, according to studies, fish disease is considered a major factor causing about 50% of overall production loss. The common outbreak and rapid spread of diseases result in large-scale fish infections in a relatively short period. This could cause massive fish death and water pollution [5]. Worse yet, it is not only parasites that can have harmful effects on human health through contact with diseased fish [6], but fish with bacteria can also infect humans with diseases such as Salmonella [7]. In addition, late identification of fish disease could result in the extinction of the whole farmed fish populations. Therefore, it is necessary to develop modern, non-destructive, fast, real-time, and automatic fish disease prediction and diagnosis techniques to keep fish healthy and safe to prevent and control disease transmission in aquaculture.

Fish disease identification is an indispensable part of modern aquaculture, and rapid and real-time diagnosis is an essential part of the early and precise treatment of diseases. However, the farmed fish will be affected by viruses, bacteria, parasites, metal pollution, and fishing damage [8]. Fish diseases are caused by a combination of different pathogens. Conventional methods involve detection via fish tissue dissection, which is destructive, time-consuming, and costly [9]. The direct diagnosis of diseased fish underwater requires a high level of technology, while the large-scale and rapid spread of fish diseases have to limit the required time for diagnosis. The diversity and heterogeneity of fish diseases increase the difficulty of diagnosis, and the accuracy of diagnosis using these various physiological indicators is low. In recent years, image-based disease-diagnosis techniques have been widely used in the diagnosis of fish diseases.

Traditional fish disease diagnosis methods are mainly based on expert systems, and great achievements have been made in fish disease diagnosis [10]. However, the diagnostic accuracy and speed highly depend on the ability and experience of the experts [11]. With the rapid development of image-processing technology, features such as texture, shape, and color from disease images could be indicators for fish disease diagnosis [12]. The use of camera images, microscopic images, spectral images, ultrasound images, and fluorescence images has been shown to provide feasibility for fish disease diagnosis, and the combination of image-processing technology and computer vision can provide non-destructive, automatic, rapid, real-time diagnosis of fish diseases, and low-cost, simple operation, high sustainability, and no pollution to water bodies [13,14,15]. However, the accuracy of disease diagnosis is generally low due to the diversity of fish species, the complexity of survival conditions, and the difficulty of obtaining high-definition images of diseased fish [16]. This has become a great challenge for accurately identifying and diagnosing fish diseases.

With improvements in automatic disease diagnosis for increasing fish production, farming efficiency and safety, and reducing the impact of diseases on humans, fish, and the environment, there have been extensive studies on fish disease identification in recent decades [5,17]. This article aims to provide a review of the techniques and methods used in recent years for the automatic identification and diagnosis of fish diseases. The application of image-processing technology in aquaculture for host and pathogen identification and diagnosis are reviewed, and indirect identification of pathogens using electrochemical genetic sensors is also discussed. Finally, the article discusses and summarizes the potential techniques for the applications of fish disease identification and diagnosis.

## 2. Image-Processing Technology

Image-processing technology has been used in aquaculture in various applications, such as fish weight/length measurement, fish classification/counting, fish detection/tracking/identification, and fish diseases [18,19,20]. However, fish disease diagnosis could be challenging because of the complexity of fish survival conditions, the diversity of fish diseases, and the heterogeneity of symptoms [16]. Image-processing technology aims to obtain higher quality and clearer images or to remove unwanted fuzzy data so that the computer can accurately diagnose and identify fish diseases [21]. The main steps encompassed by image-processing technology are image acquisition, image pre-processing, image segmentation, feature extraction, target detection and recognition, and classification [22], as shown in Figure 1.

### 2.1. Image Acquisition

The image quality directly determines the accuracy of fish disease detection based on the image. The detection of disease using images includes two steps: the collection of high-quality images and the pre-processing of the acquired images. There are many different methods for acquiring images of fish in the aquaculture field, such as camera, microscopic, spectral, fluorescence, and ultrasonic imaging. Fish disease is attributed to the host, pathogen, and water environment, which contribute to disease in fish, as shown in Figure 2; thus, it could be diagnosed directly or indirectly via image acquisition in these three directions. According to the different acquisition methods, fish disease image acquisition can be divided into two categories; one is in the aquatic natural environment and the other is under external human intervention. Image acquisition in the aquatic natural environment is not affected by human interference, and these images are mainly used to detect diseases in fish by observing abnormal fish surfaces or abnormal fish swimming; the images mainly include: camera images, spectral images, and ultrasonic images. Image acquisition under external human intervention can cause damage to the fish, and this category mainly includes: expert systems, microscope images, and fluorescence images.

Currently, most aquatic fish disease diagnoses are primarily based on 2D images. However, 2D images cannot effectively solve the problems of occlusion, overlap, and high numbers in diagnosing fish diseases in water. In contrast, 3D images could potentially overcome these problems to some extent [23]. For example, in the detection of changes in fish behavior caused by disease, 2D images cannot accurately determine the speed of fish movement and the spatial location of the fish, while using 3D images could accelerate the fish body and the spatial state of the fish [24]. In the future, 3D images could provide new methods for the timely detection and diagnosis of fish diseases from different perspectives.

### 2.2. Image Pre-Processing

Image pre-processing is an key process in fish detection [25], to remove noise or eliminate unnecessary information in the image and improve the quality of the image after acquisition. Image segmentation, detection, recognition, and classification in the process of fish disease diagnosis could be enhanced by image pre-processing, thus improving the accuracy of fish disease diagnosis. Additionally, the accuracy of fish disease detection could be improved by high-quality images. The following section describes related methods of image pre-processing in aquaculture, such as denoising, sharpening, smoothing, and enhancement.

#### 2.2.1. Image Denoising

The presence of noise interference affects not only the image quality, leading to blurred image edges, but also the image segmentation, detection, feature extraction, etc. [26], reducing the accuracy of diagnosis. To improve the accuracy of fish disease detection, the quality of the images needs to be improved. However, various disturbances are inevitable in the process of image acquisition and storage [27]. Noise reduction is therefore essential before edge detection, image segmentation, target detection, and feature extraction of the images. Denoising uses linear or nonlinear filters to reduce various instances of noise in the image. Most of the existing noise-reduction methods use wavelets and filters [28]. To eliminate noise, the traditional linear filters cause fuzzy edges in the image while eliminating noise, consequently leading to information loss. To overcome the drawbacks of conventional linear filters, noise reduction was performed via Haar wavelet transform, which was then combined with histogram equalization to improve the image quality [29]. The experimental results showed that the method had a strong denoising ability. For underwater images with problems such as low contrast and poor color balance, the general approach cannot be directly adapted to denoising. A novel and improved median filter was used to reduce noise in fish images, capable of handling pixels contaminated by impulse noise [26]. This could reduce image noise and overcome the shortcomings of standard median filtering.

Although many image noise-reduction algorithms exist, some only consider the scattering component without color correction. Creating an algorithm that could simultaneously address noise, light absorption and scattering effects remains a big challenge. To improve image quality and reduce noise levels, intelligent image-noise reduction could combine artificial intelligence and decision-support techniques to make intelligent choices for image processing to meet the needs of a variety of researchers.

#### 2.2.2. Image Sharpening

The sharpening of images is required to enhance the edges and contours of the fish images in detection. The purpose of sharpening (high-pass filtering) is to compensate for the contours of the image so that the edges of the image become clearer. The wavelet-based panoramic sharpening Landsdt7 image sharpening algorithm could improve the spatial resolution of multispectral imaging and retain useful information [30].

Many different methods have been proposed for image sharpening, such as histogram equalization, Laplacian sharpening, and filtering methods in the transform domain. However, one of the main challenge for these methods is avoiding enhancement noise and ringing effects along the edges in image sharpening. Future methods of image sharpening need to be improved to address the problem, while also making the image edges and contours sharper.

#### 2.2.3. Image Smoothing

Image smoothing is quite different from image sharpening. Smoothing (also known as low-pass filtering) makes the image brightness smooth and gradual, lessens the mutation gradient, and improves the image quality. However, the smoothing process of the image could lead to a fuzzy edge. When the fish disease detection information is not on the edge, the image could be smoothed to eliminate the interference of other information. In processing echo images of fish, Shigeyasu and Ogasawara utilized the two-dimensional exponential smoothing method to smooth the echo images [31], which could achieve the effect of traditional mask smoothing. Smoothing could prepare the image for the extraction of features at different scales to improve the subsequent detection accuracy. Multispectral images were smoothed using the Savitzky–Golay smoothing algorithm [32], and the skin color features on the body surfaces of fish were extracted. The experimental results indicated that the classification accuracy of live fish was up to 84%. The shapes and proportions of features are not considered in traditional smoothing, and the application of morphological technology could solve these problems. Chai et al. proposed a method for smoothing images using multi-scale morphology [33]. By employing feature spaces of different scales to smooth noise, the experimental results proved to be better than other traditional smoothing algorithms. The future development of image smoothing should be in the direction of simple and fast processing.

#### 2.2.4. Image Enhancement

Fish disease diagnosis is primarily based on the color and characteristics in the image to target the infected area. Lower quality from fuzzy images reduces the benefits of image segmentation and target detection. Hence, the accuracy of diagnosis declines significantly. Image enhancement could improve the image’s contrast and solve the color scattering and projection problems [34]. In recirculating water aquaculture, the water system causes light scattering, resulting in insufficient light. Traditional image enhancement may not be appropriate for aquaculture environments [35]. A method to enhance images is using the MSR algorithm combined with a grayscale nonlinear transformation [36], which used MSR to improve the contrast of images and effectively reduced the effects of low illumination and lighting inhomogeneity. The experimental results showed that this method could enhance the brightness of the image better than other typical enhancements. However, the underwater environment is complicated, and the haze effect obviously influences the image quality. The image dehazing and wavelength compensation (WCID) method used the pixel threshold to stretch pixels to remove underwater smog, and the pixel threshold, via backpropagation, to achieve image enhancement [37]. This simulation resulted in WCID better enhancing the image but was not used in a real underwater environment. Acquiring images of moving fish may also lead to fuzzy images, so an underwater image-enhancement model was used for color enhancement and color correction of the images [28]. This model was also flexible in adjusting the color appearance of underwater images.

Image enhancement enables researchers to overcome the challenges of specific underwater environments, such as blurred images and fish distortion. Neural networks successfully handle noisy, faint, and blurred underwater images. In the future, the algorithms for image enhancement and complementary information-fusion techniques will be optimized to produce new images with higher resolution, thus improving the information in the images.

### 2.3. Image Segmentation

Every image carries uninteresting information. To obtain a region of interest in the image, accurate segmentation of the image is required. Image segmentation also plays an important role in detecting and recognizing targets. Since the lesions are presented locally, image segmentation is required to improve the recognition of fish diseases.

#### 2.3.1. Image Segmentation for Computer Vision

The development of computer vision-based segmentation in recent years has significantly improved segmentation performance by reducing human involvement and automating image segmentation. A method for fully automated fish gill segmentation based on computer vision was employed in [38], which used an active contour method to precisely segment the pixels in the gill region of the image and used LAB color space to improve the segmentation accuracy. Authors have also used the color threshold algorithm with removal of the background and deep learning segmentation in fish body measurements of tilapia [39]. The color threshold algorithm was used to generate a segmentation map that only contains the fish, and after that, the deep learning algorithm was used to segment the fish body and fin, and the method was able to distinguish the fish body and fin successfully. In overlap segmentation of fish in images, traditional segmentation algorithms can lead to serious segmentation errors. To reduce the segmentation error of overlap in the image, CNN segmentation and gradient refinement techniques successfully processed images of overlapping schools of fish for individual fish detection [40].

#### 2.3.2. Traditional Image Segmentation

Image segmentation technology includes not only machine vision and deep learning methods but also other methods, such as segmentation via K-means clustering, segmentation via thresholds, contour-based segmentation, etc. Common clustering takes the principle of uniformity based on the region method, where there is only one pixel of a feature in each region. Image segmentation based on the threshold method needs to select the optimal threshold value for image segmentation. The active contour segmentation method provides an ideal framework for innovation in segmentation. Table 1 below shows the application of the clustering method, threshold method, and contour segmentation on an aquatic environment.

Image segmentation involves the consideration of features such as the shape, color, texture, and intensity of the object. Traditional segmentation and machine vision segmentation still need to be improved in terms of segmentation accuracy and efficiency. In the future, spatial and temporal information fusion could be used to automatically quantify and segment, and to solve the effects caused by, for example, shadows and blurred backgrounds.

### 2.4. Feature Extraction

Feature extraction refers to the use of a computer to extract image information from an image, where the features are the same type of measurement data in the image. Different features could be used for the automatic and accurate detection of fish diseases. Common feature extraction can be divided into a color feature, texture feature, shape feature, and spatial feature. The more prevalent ones are the Haar feature, LBP feature, HOG feature, Shif feature, etc. Different features have to be selected when studying different targets, while the accuracy of single-feature extraction and the accuracy of manual feature screening by artificial neural networks are no longer sufficient for fish disease detection in aquaculture. Table 2 below describes the application of feature extraction in aquaculture fish detection. With the development of computer vision, deep learning has been greatly applied in the direction of automatic feature extraction, which has greatly improved accuracy.

The accuracy of detection could be improved by dimensionality reduction of the extracted feature data. Image information could be downscaled by projecting it into a lower-dimensional space using a linear transformation [55]. Beyan et al. used principal component analysis for feature processing in trajectories in their approach to analyzing anomalous fish trajectories and obtained an accuracy of 71% [56]. However, single-feature acquisition may not yield the ideal result. Li et al. employed principal component analysis to reduce the dimensionality of 11 feature parameters [57]. Four principal components were selected to classify fish and the average recognition rate was 99.67%.

Feature extraction is not efficient on large datasets under both constrained and unconstrained conditions, and its accuracy is unsatisfactory. For image-processing technology in dynamic conditions, further research is needed to extract different features of images, for example, physical appearance, motion, and sensory cues. Association of the selected features with the assigned categories could be further analyzed and compared with other advanced models such as deep learning, reinforcement learning, generative adversarial networks, etc., and further output to microcontrollers; this could improve accurate, efficient, and robust recognition in the dataset. In the future, there is also a need to find a way to reduce central-processor memory consumption.

### 2.5. Target Detection

The purpose of target detection is to detect and localize the object of study in the image or video. Many fish diseases cause behavior changes in fish, and target detection could detect and track fish movement [58]. The detection and tracking of behavior would be available to provide a new and viable method for the rapid detection of fish diseases. Nowadays, research on target detection is little-applied in fish disease detection; rather, it is mainly focused on fish movement tracking. Traditional Gaussian mixture models have difficulties in maintaining and updating motion detection. An improved Gaussian mixture model was introduced to detect moving fish targets [59]. The model, combined with the Canny edge operator, could measure the moving edges more effectively and conveniently. It had higher stability and adaptability than the traditional algorithm for fish motion detection. The head of a fish is a distinctive feature in fish appearance, and the tracking of fish can be carried out with the help of the head. Qian et al. used fish heads to locate and estimate the direction of motion [60] and used the cost functions to correlate the targets between consecutive frames. The experimental results showed that multiple fish could be tracked simultaneously using fish head features. The accuracy of tracking could be significantly improved using machine vision. Wang et al. used CNN to identify the fish head for tracking. The method used scale space to detect fish heads and tested five different fish counts [61].The results indicated that head-based CNN tracking was not subject to frequent occlusion. Target detection results based on a single feature may be inaccurate for the resulting detection, and multiple features can be used for target detection.

Fish are usually infected by infectious fish diseases in a short time. To diagnose fish diseases quickly, several fish need to be followed up and detected at the same time. Simultaneous diagnosis would require simultaneous tracking of multiple fish, and Lan er al. proposed a method for the fast detection and identification of target motion that used a combination of three-frame difference and background subtraction to detect motion and obtain motion information [62]. This method was able to detect fish movement accurately, completely, and quickly in the experiment. When multiple fish were tested, there would inevitably be no occlusion, overlap, etc. To solve the occlusion problem, Zhao et al. proposed a multi-target tracking algorithm. It first used a Kalman filter to point estimate the fish motion information [63], and then, explored the inter-frame relationship matrix to solve the occlusion problem. They found that the algorithm would effectively improve the performance and robustness of occlusion tracking.

Comparing the deep neural network of CNN with the shallow neural network of RNN, CNN achieves higher classification accuracy in target detection and allows real-time implementation while being less time-consuming. Using CNN, this method overcomes the trouble of distortion correction at traditional locations without correcting distorted images. In the future, continuous stability and faster detection will be necessary for the real-time detection of targets. The speed and accuracy of detection to improve the performance of detection could be accelerated and based on data association by introducing target-region and motion-direction features to solve the occlusion problem.

### 2.6. Target Recognition

The purpose of target recognition is to detect the research object in the image or classify a target of the same type in the image. Target recognition is mainly applied in aquaculture for fish classification and identification, and fish state analysis. In most studies, fish can be recognized using extracted features such as color, morphology, and texture. Sun et al. proposed an automatic recognition algorithm based on color and morphology [64], which integrated a weighted histogram of the target model and was able to enhance the recognition of targets while also allowing real-time localization of underwater robots. Traditional algorithms have improved the accuracy, efficiency, and scope of recognition. However, insufficient underwater light, complex environments, and low contrast cause great difficulties in fish recognition in underwater images using conventional algorithms. In recent years, with the continuous improvements in convolutional neural networks, many researchers have greatly improved the accuracy of fish classification using convolutional neural networks. Ben et al. proposed the use of a convolutional neural network, AlexNet, with transfer learning for automatic fish classification and used the AlexNet network to obtain features from images to identify fish [65], which was experimentally proven to be able to reach 99.45% accuracy on real data. To improve its performance, Ben et al. conducted fine-tuning of AlexNet on the dataset, and finally, mixed the feature extraction and fine-tuning methods to improve the performance of this method [66]. Deep et al. proposed a hybrid convolutional neural network framework that uses vector machines (SVM) and K-nearest neighbors for classification [67], respectively. The framework was experimentally superior to existing underwater recognition in terms of accuracy, precision, recall, and f-score, and gave an accuracy of 98.79%.

The performance of target recognition has been improved using techniques from image processing as well as deep learning, but there are still potential studies we could perform to improve the performance. Underwater recognition is inevitably susceptible to transient-object blur, illumination, and shadow changes or drift, all of which could lead to recognition errors. These issues, as well as computational acceleration, could be addressed in the future. In future work, accuracy could be improved by developing better detection systems that include more sensitive cameras and better illumination. The performance of the system could be further improved by integrating temporal information from underwater video data to estimate the different poses of the fish.

### 2.7. Classification

Classification is based on the characteristics of the image to discriminate the type, and the classifier is used to learn the features using a training set to classify the fish in the image. Due to the heterogeneity of fish diseases, it is necessary to classify fish during fish disease diagnosis to improve the accuracy of diagnosis. The commonly used machine learning classifiers are SVM, MLP, neural network, decision tree, random forest, and naive Bayes. Table 3 shows the classification accuracy of various classifiers in the experiments. Saberioon et al. conducted a comparison of four different classifiers based on image features [68], and the experimental results exhibited that the Support Vector Machine (SVM) had the highest correct classification rate at 82%, while K-Nearest Neighbor (K-NN) was the least accurate (40%).

## 3. Intelligent Diagnosis Method of Fish Diseases Based on Images

Fish diseases are caused not only by common pathogens such as parasites, fungi, bacteria, and viruses but also by some common physical phenomena and chemical agents that could cause fish diseases. Changes in the water environment could also cause stress to fish health. Fish diseases cause surface and behavioral as well as internal tissue changes, as shown in Figure 3 below. Images can be used in aquaculture to diagnose fish diseases by detecting changes in hosts, pathogens, and the environment. Most of the existing studies have shown that the detection of hosts is mainly based on changes occurring on the surface of the fish, and the detection of pathogens is mainly based on changes in the internal tissues of the fish, such as deformation, necrosis, decay, and bleeding. However, the methods of detecting pathogens can only roughly detect the existence of pathogens. There are many different types of pathogens in fish and the identification of pathogens is usually dependent on the biochemical pathogen-detection methods. Accurate detection of diseases must be performed with the help of high-quality images. Firstly, image-processing technology is used to improve the quality of fish images; then, high-quality images are used for segmentation, detection, and identification. Finally, fish diseases are diagnosed based on fish classification.

Paragraph 3 describes fish disease expert systems and camera images for fish disease diagnosis based on changes in the body surface of the fish; pathogen detection using microscopic images, spectral images, ultrasonic images, and fluorescence images; and finally, the detection of pathogens in the water column based on electrochemical genetic sensors. These methods can be divided into two groups according to whether the detection method is damaging to the fish. The damaging ones include expert systems, microscopic images, fluorescence images, and ultrasonic images; the nondamaging ones include camera images, spectral images, and indirect detection by sensors. Additionally, the detection methods of live or dead fish can be divided into two groups; the detection of live fish includes the use of camera images, spectral images, fluorescence images, ultrasonic images, and indirect detection by sensors; the detection of dead fish includes expert systems and microscope images.

This is because camera images and spectral images can be used to detect fish diseases using images taken in the water, so they are mainly used for live fish disease detection, and they do not cause damage to the fish’s body. The expert system needs to detect fish diseases through eight parts: the body surface, head, gills, abdomen, scales, fins, muscles, and internal organs. Therefore, the expert system is mainly applied to dead fish disease detection, and it causes damage to the fish body. Microscope images need to dissect the fish body to obtain internal tissue images to detect diseases, so microscope images are mainly applied to dead fish and cause damage to the fish body. Fluorescence images are used to detect live fish because the fluorescent agent in the water flows into the tissue of the fish, but the fluorescent agent inside the fish can cause some damage to the fish. The use of ultrasonic images is mainly applied to the detection of disease in live fish underwater in a complex environment, but the wavelength of ultrasonic waves can cause damage to the fish’s body.

### 3.1. Aquaculture Expert Systems

Computers combined with expert knowledge of fish diseases constitute an early expert system commonly used for fish disease diagnosis in aquaculture environments. The basic structure of the expert system is composed of a knowledge base, a database, a reasoning machine, a user interface, and an expert learning module. The system is a way to imitate a human expert to make decisions on detection results, as shown in Figure 4 below. Zhang et al. developed a fish disease diagnosis system that contains a large number of fish disease data and images and could simulate a large amount of specialized fish disease knowledge [73]. The system could diagnose 126 fish diseases through a user interface. The expert system could improve the accuracy of fish disease detection with a large number of data and expert experience, but it did not consider the complexity of expertise that might be required in the detection method or the problem of remote diagnosis. To reduce the involvement of fish disease professionals and increase the accuracy of diagnosis, Lou et al. proposed a case-based reasoning system based on image retrieval [12]. The system could search and match images in the database according to color, morphology, texture, and other features. The accuracy of detection could be significantly improved by using a multi-feature search.

To meet the needs of aquarists in different situations and to solve the problem of diagnostic delays due to distance. Zhang and Li proposed a call-center-oriented consultant system for fish disease diagnosis in China [74], which was combined with a fish disease diagnosis system to provide remote fish disease diagnosis for rural areas with slow Internet transmission. Compared with the slower transmission speed of the Internet, the use of mobile phones could increase the speed of information transmission and is more convenient. Wang and Li proposed a fish disease diagnosis system using a short message service (SMS) via mobile phones [75], which achieved rapid and long-distance diagnosis through the automatic exchange of SMS and data information. To further improve the integration efficiency of the fish disease diagnosis system, Ma et al. proposed the development of a multi-agent fish disease diagnosis system architecture [76], which was based on the advantages of multi-agent distributed computing to solve the integration problem of the system, reduce the burden of centralized processing, and decrease the data traffic and network load. The development of smartphones has improved the quality and speed of image acquisition to diagnose aquaculture diseases faster by improving the quality and speed of image acquisition. Sun and Li developed an aquatic animal disease diagnosis system using an expert system combined with an Android system [77]. Combined with the traditional expert system, it was faster in terms of detection speed. The real situation of the fish disease site and the degree of water pollution cannot be transmitted to experts through mobile phones, and the use of aquaculture surveillance video could overcome these problems to a certain extent. Ma et al. proposed a remote fish disease video-based diagnosis expert system [78], which used remote video to provide experts with information on the aquaculture environment and dynamic fish conditions, and realized accurate and timely diagnosis of fish diseases by remote experts through direct communication between field personnel and experts.

Fish disease diagnosis data for aquaculture are ambiguous, random, and uncertain. To eliminate incomplete and uncertain information in fish disease diagnosis and to simplify the attributes to be identified, Yuan et al. combined expert systems and fuzzy technology to improve the flexibility of reasoning using the strategy of inference modeling [79]. Rough set theory is a mathematical tool for dealing with fuzzy and inaccurate knowledge. Expert systems combine rough sets and fuzzy clustering algorithms to handle large amounts of data and uncertain information, as well as to eliminate redundant information [80,81,82]. Traditional expert systems are less intelligent due to complex knowledge acquisition and inefficient reasoning. To combat the disadvantages of a low level of diagnostic intelligence and poor practicality of the expert system, Deng et al. proposed a new neural network-based expert system for fish disease diagnosis [83]. The system simplified the knowledge acquisition of traditional expert systems and had strong adaptability and fault tolerance for uncertain and incomplete knowledge.

Expert systems rely on the knowledge and experience of fish disease experts to logically identify and diagnose diseases. There is no shared database among existing expert systems, reasoning about new fish diseases can only be analyzed with a strong background of knowledge, and the accuracy of diagnosis is low. Expert systems require the intervention of staff and fish disease specialists and may work as a semi-automated diagnosis. Fish disease outbreaks and infections can be spread in a relatively short time, and this semi-automated model is no longer relevant in the face of today’s large-scale aquaculture. In the future, all fish disease expert systems could be networked to share the database of various fish diseases and combine the latest 3D images to build AR models; in this way, experts could understand the conditions of diseased fish more clearly to make a more accurate and timely diagnosis.

### 3.2. Diagnosis Based on Body Surface Images

Nowadays, fast, real-time, and automatic diagnosis of diseases is urgently needed, and image-processing technology combined with computer vision has provided automatic and fast diagnosis of fish diseases. This diagnosis has a faster speed compared to expert systems and is more effective in the correct diagnosis of new diseases. In general, fish diseases can be roughly divided into infectious and non-infectious, with the former caused by viruses, bacteria, fungi, and parasites, and the latter caused by environmental stress, genetic factors, and nutritional deficiencies [84]. The current methods of disease detection are still mainly reliant on changes in the surface and internal tissues of the fish. In contrast, behavioral detection may provide a faster and more feasible method of fish disease diagnosis. Fish diseases in water plant culture could be detected by both in-water images and flat images simultaneously, as shown in Figure 5 below.

#### 3.2.1. Diagnosis Based on Camera Images

To overcome the limitations of expert system diagnosis and to improve the accuracy of fish disease diagnosis, the development of image processing and machine vision has provided methods for the automatic detection of fish diseases. The surface of the fish is considered to provide the key information about the disease infection, and camera images could use the visible changes produced on the surface of the fish as a basis for rapid diagnosis. Hu et al. designed a computer vision-based system for identifying infected fish [85]. It used color and texture as fish classification features to accurately identify Chinese carp diseases through image analysis. The disease could lead to infected areas on the body surface that are very different from the normal body surface, and the features of the infected areas could be used to make a quick diagnosis of the fish disease. Lyubchenko et al. used digital image-processing methods to conclude that it is feasible to detect infected areas in fish and diagnose fish diseases [86]. The diversity and heterogeneity of fish diseases increase the difficulty of diagnosis, and multi-feature extraction could improve the accuracy of diagnosis to a certain extent. Malik et al. used HOG and two feature descriptors to detect fish diseases and extracted features using fast segmentation test features and directional gradient histogram feature descriptors to identify the region of interest of EUS-diseased fish [87]. To enhance the efficiency of the present fish disease image recognition technology, Chakravorty proposed a new method of detecting fish disease via smartphone based on the augmented reality and image-processing techniques [88]. The images were taken using an AR mobile application, and then, a combination of enhancement modeling and image-processing technology was used to identify the ulcerative syndrome (EUS) in the images. The experimental results showed that this method was effective for disease diagnosis. This experiment was a step forward in improving the method of fish disease identification and prevention.

Feature extraction of the region of interest could directly affect the accuracy of automatic fish disease diagnosis. The more prominent the fish disease characteristics acquired, the more accurate the diagnosis of the fish disease. High-dimensional fish disease data could make pattern recognition very difficult, and dimensional processing of the extracted features using principal component analysis (PCA) could improve the accuracy of disease detection. Chakravorty et al. segmented diseased areas of fish images based on color features and K-means clustering and identified diseased-fish images using principal component analysis [89]. The experimental results showed that image-processing technology could identify diseases with higher accuracy and calculate disease areas correctly. Malik et al. used image-processing technology to identify and recognize EUS disease in fish and applied principal component analysis to the extracted features in the experiment. Their experimental results showed that this technique has a higher accuracy rate [87]. Maylawati et al. utilized mixed principal component analysis and K-nearest neighbor (KNN) to detect fish diseases [90]. The results demonstrated that principal component analysis and KNN could be used to classify catfish diseases with a detection accuracy of 90% or less.

Neural networks provide a feasible method for rapidly and accurately diagnosing fish diseases. Lopes et al. evaluated artificial neural networks for fish disease detection and concluded that artificial neural networks were applicable for disease diagnosis when the database was sufficient [91]. Neural networks provided feasibility of the automatic identification of fish diseases in real-time and also had a high accuracy rate. Scholar et al. proposed a real-time fish disease diagnosis system with high accuracy [92]. The system used principal component analysis and accelerated segment test feature detectors of neural networks to detect the disease in a shorter time. Experiments showed that the FAST-PCA-NN combination had better classification accuracy than the HOG-PCA-NN combination, and the accuracy reached 86.0%.

The diversity of fish diseases and the heterogeneity of symptoms increase the difficulty of detection. To study the interaction of multiple symptoms and draw a conclusion, Cornelia proposed an intelligent aquatic decision-support system model (IDSS) [93]. This system aimed to develop new methods and technology for the rapid diagnosis, treatment, and prevention of fish diseases caused by parasites. When different diseases appear in fish, it is necessary to evaluate and classify the disease along with the identification of the disease. Sucipto et al. compared the classification algorithms with the disease data of catfish and carp [94], and the results showed that the C4.5 algorithm could be used to evaluate fish disease performance. Waleed et al. proposed automatically identifying three different types of fish diseases [95], and used different CNN architectures according to the different color spaces on the dataset images. The AlexNet architecture achieved superior results in the XYZ color space.

Image-processing technology combined with machine vision could provide automatic, accurate, and real-time diagnosis of fish diseases. Diagnosis using this technology also has the advantages of high speed, strong robustness, and easy implementation. However, image diagnosis does not provide much information about all the features shown in the disease images, and the accuracy of diagnosis needs to be improved. In the future, deeper convolutional neural networks could be used to acquire disease features at high latitudes in images, and obvious visual differences arising from disease images in terms of morphology, contour, color, location, texture, etc. could be obtained for diagnosis. To solve the problems of fewer image datasets and higher acquisition costs, transfer learning could be used to reduce the training dataset and accelerate the implementation of an image diagnosis system.

#### 3.2.2. Surface Damage Detection

Fish are often subjected to a variety of physical, chemical, and biological stresses that could damage the surface of the fish. This could be harmful to the health of fish and lead to fish diseases [96]. Adamek et al. used computer-aided image analysis to assess the extent of surface damage in fish [97]. The experiment was extended to assess epidermal contusions up and down skin tissue in six pond fish species to derive the frequency of injury occurrence. To detect the damage degree of the whole fish, Tran et al. used the ZED stereo camera to determine the damage rate of the fish surface based on the fuzzy C-means clustering algorithm and L*a*b* color space [98]. The method took advantage of the L*a*b* color space and the fuzzy C-means clustering algorithm and used median, bilateral and Gaussian filters to improve the accuracy; the experimentally measured damage rates were close to the true damage rates. Uhlmann et al. developed a method and device to automatically calculate the percentage of the visible bleeding-damage surface area in whole fish using high-resolution images [99]. The experiment segmented the fish into head, body, and neck regions to detect injury in each region, and the definition of the head region resulted in overall automatic scores that were smaller than those of the raters. However, the experiment was based on spot size to identify bruises and spot hemorrhages, which had not been well defined.

The health and defects of fish fillets could reflect disease problems in aquaculture fish and affect the benefits of aquaculture. Balaban et al. developed an image-analysis method for the quantification of gaping, bruising [100], and blood spots in sockeye salmon (*Oncorhynchus nerka*) fillets. The method adaptively applied thresholds to the images based on the average color of the rounded corners, allowing the automatic detection and quantification of defects in salmon fillets. However, there were differences in the calibration colors of the rounded corners taken by this method using two different cameras. Xu and Sun developed an automated imaging analysis method based on convolutional neural network features to detect split fillets [101]. The experimental results showed correct classifications of 0.927 and 0.916 on the cross-validation and test datasets, respectively.

#### 3.2.3. Pollution Detection

Changes in the aquaculture environment could have an impact on fish health, and water pollution could lead to significant changes in not only fish behavior but also fish eyes. Eguiraun et al. developed a non-invasive method to analyze the non-linear trajectory of fish responses to random events, which was suitable for identifying changes in fish event responses [102]. The detection of fish eye characteristics could reflect problems with fish health and water pollution. The use of computer vision to extract different features of fish eye tissue regions allows for the rapid detection of fish ill-health due to water pollution. Sengar et al. provided a non-destructive computer-aided method to identify fish after pesticide exposure [103]. They selected different features from fish eye tissue and compared the accuracy of different classifiers. The highest recognition accuracy of 96.87% was achieved using random forest. Issac et al. proposed an automated non-destructive image processing method for identifying visual changes to distinguish between controlled (untreated) and heavy-metal-exposed (treated) fish [104]. The method used computer vision to detect changes in the eyes of fish caused by heavy metals, and the experiment allowed for the rapid identification of contaminated fish. The experimental results showed that similar results could be obtained for fish contaminated with different heavy-metal exposures.

### 3.3. Diagnosis of Internal Tissues Based on Microscopic Images

Fish diseases can be detected not only from surface changes, but also through the detection of pathogens in fish body tissues to determine the disease, and the detection of pathogens can be completed with the help of microscopic images. Park et al. developed a fish disease diagnosis system based on microscopic-image processing [11]. The system extracted pathogen areas from microscopic images and compared infected tissues with an existing database of pathogens through image-processing technology to determine the final diagnosis. The experimental results showed that the system was more convenient, more consistent, and faster than general diagnostic methods. Ceong et al. developed an image-based rapid diagnostic system that could diagnose parasites using microscopic images or videos [105]. López-Cortés et al. used new technologies of mass spectrometry and machine learning and combined them in an automated platform to perform rapid detection of pathogens in salmon farming to obtain specificity and susceptibility patterns for each pathogen [106]. The experimental results showed that the platform had good performance with accuracy, sensitivity, and specificity greater than or equal to 92%

The combination of microscopic images and machine vision improves the accuracy of pathogen detection. Coelho et al. proposed a machine vision system for the automatic detection of parasites [107]. The accuracy of the system was different in laboratory and industrial settings, with laboratory samples reaching 98% classification accuracy, compared to 73% classification data for the industrial samples. Mohamed and Adl developed a neural network-based fish disease diagnosis system that extracted features from microscopic images via the ORB feature-extraction technique, and then, used decision trees for classification [108]. The experimental results showed an accuracy of 87.5% in classifying fish diseases. The system was a fast and effective method of identifying and detecting *Trichodina* and *Gyrodactylus* diseases.

The rapid diagnosis of multiple fish diseases requires the classification of pathogens and could provide the methods needed for fish disease treatment. Elham Yousef et al. proposed an approach to identifying Monogeneans (Platyhelminthes) by using digital image processing and K-nearest neighbors (KNN), and the overall classification accuracy of this method was 90% [109]. Zhan et al. proposed a parasite classification method based on digital image-processing technology, which first extracted the edges of spores and cysts [110], and then, used the Canny algorithm in image-processing technology, ellipse fitting, and the improved region-growing algorithm to achieve classification. However, the accuracy of this classification method was related to image clarity and could not identify unclear sporozoites.

Pathogens are ubiquitous in aquaculture and can be tested with the advantages of zebrafish to describe and determine the pathophysiology and mode of infection. In recent years, the image analysis and quantification of zebrafish have focused on phenotypic assessment, neuronal-structure quantification, vascular-structure reconstruction, and behavioral monitoring [111], as shown in Table 4 below. Since the embryos of zebrafish are completely transparent, they provide the ability to observe changes in cells within the embryo. The study of zebrafish diseases not only helps people to understand the changes inside the fish but also allows the screening of drugs.

Microscopic images enable the effective detection of pathogens in the host. However, the precise dissection of fish tissues requires specialized personnel, and the acquisition of microscope images requires the expertise of the researcher. Although microscopic-image diagnosis could improve the accuracy of the diagnosis of pathogens, the professional operation is not suitable for large-scale aquaculture, and obtaining microscopic images can also cause trauma to fish. In the future, an electron microscope could be connected to an LCD screen to automatically acquire images of fish diseases and reduce the operational difficulty of microscope use. Additionally, wireless transmission could be used to directly connect the collected images with the computer to increase the speed of the microscope’s detection of pathogens.

### 3.4. Pathogen Detection Based on Spectral Images

An automatic fish disease detection system combining image-processing technology with machine vision is used as an automatic, efficient, and non-invasive method for disease identification, allowing the effective assessment of disease with the help of visible changes occurring on the surface of the fish. In addition to the ability to use microscopic images, spectral imaging technology could also be used for the detection of pathogens. Spectroscopy technology is used in some aquaculture facilities with poor lighting and turbid water, and hyperspectral imaging can also provide both target-object and spatial information. Additionally, this technology is less costly and is easy to develop. Near-infrared spectroscopy had been used for a long time to detect fish damage [119]. According to Lin’s study, the non-destructive detection of fish skin and scales using visible light produced a relative error of 0.83% in the experiment. This study showed that visible light and short-wave near-infrared techniques are able to detect fish abrasion defects.

It is estimated that as many as 30,000 species of parasite use fish as intermediate hosts [120], which poses a significant threat to the health of fish. Hyperspectral imaging, as an intelligent and non-destructive technique, could provide an intuitive and simple approach when studying infected fish tissue. Sivertsen et al. developed a hyperspectral imaging system to detect nematodes in cod fillets under industrial conditions [121]. However, the total experimental detection rate of nematodes was relatively low (only 58%). To improve the accuracy of detection, Sivertsen et al. improved the developed hyperspectral imaging system [122]. This study utilized a local calibration method and classified dark and light colors using a Gaussian maximum-likelihood classifier. The experimental classification probabilities for dark and light colors were 70.8% and 60.3%, respectively, and the performance of this system at the industrial level was already comparable to that of manual inspection. Hyperspectral imaging can directly detect fish parasites in complex environment underwater. Pettersen et al. detected and classified sea lice (*Lepoptherius salmonis*) using underwater hyperspectral imaging [123]. In the study, the UHI technology was used to detect and classify different stages of sea lice based on differences in the spectral characteristics of sea lice. The average success rate of the experimental detection was 82%.

Bacteria, as pathogens, can cause lesions and lead to tissue mutations in fish. Wortberg et al. used biochemical methods with Fourier transform infrared spectroscopy (FT-IR) combined with artificial neural network analysis to characterize the strains, while DNA sequencing was performed on some strains using biochemical methods. The results finally confirmed that the differentiation method with the FT-IR module was 97.4% correct [124]. However, this study utilized a complex and time-consuming biochemical technology that increased the difficulty of the experiment. For the rapid and real-time detection of bacteria in fish, He et al. constructed a new optimized model using near-infrared hyper-spectroscopy combined with chemometric analysis [125]. This model was calibrated using the least squares support vector machine (LS-SVM) algorithm for the spectral data, and the experiments showed that hyperspectral values of 900–1700 nm had great potential for the detection of lactic acid bacteria. To improve the spectral and spatial information in the region of interest of the spectral images, He and Sun evaluated the selection of spectral wavelengths for the bacterial contamination of salmon meat [126]. Cheng and Sun developed a 400–1000 nm spectrum for the detection of E. coli in grass carp [127]. The experiment showed that the simplified multilayer linear regression model yielded better predictability than PLSR.

Viruses are non-cellular organisms, and each virus contains only one type of nucleus—either DNA or RNA. Spectral images have been confirmed for the detection of viruses, but the accurate detection of fish viruses is now mainly performed via PCR. Virus detection based on spectral images has been applied to botany, and good progress has been made. For example, Morellos et al. used spectroscopy to obtain spectral images of healthy and diseased tomato leaves, and then, used the XY-fusion network (XY-F) and the multilayer perceptron with automated relevance determination (MLP-ARD) to estimate disease presence and virus load, respectively, in tomato leaves [128]. The results showed that the MLP-ARD-classified machine outperformed the XY-F network overall, with an overall accuracy of 92.1%, compared to 88.3% for XY-F. In the future, the quality of the spectral images will be improved by optimizing the fish habitat, lowering the impact of the complex environment in the spectral images, and improving the quality of the spectral images through suitable image-processing technology. In this way, better spectral images can be obtained and used for the detection of fish viruses.

Spectral images can detect pathogens in fish tissues quickly, noninvasively, and effectively, but spectral images mainly detect pathogens in processed fish tissues. There are fewer studies on disease detection in live fish. Spectral instruments are less adaptable when detecting fish diseases in wet aquaculture environments. In subsequent research, the hyperspectral image could be miniaturized and could be automatically adapted to the underwater environment. Hyperspectral imaging advantages are combined with behavioral detection for the rapid diagnosis of fish diseases in aquaculture.

### 3.5. Parasite Diagnosis Based on Ultrasonic Images

Ultrasound does not require the processing of the fish body since it can detect fish diseases in water easily, especially problems inside the fish body. The use of ultrasound for the detection of parasites in fish has been studied for decades. Freese proposed the early use of ultrasound for the detection of trichinella in whitefish [129]. This study used ultrasound echoes of the parasite to localize and study the three-dimensional distribution of the parasite in the fish. Ultrasound not only locates the parasite but also assists in the treatment of fish diseases by inhibiting the parasite. Navot et al. studied the effect of ultrasound on the immunization of goldfish (*Carassius auratus*) [130]. The experiment used a control group, one group undergoing ultrasound without being injected with reagents, and another group injected with reagents without undergoing ultrasound. The experiment revealed that ultrasound had an advantage in fish vaccines. Different wavelengths change the behavior of the parasite, but different ultrasonic wavelengths could also be harmful to fish health. Skjelvareid et al. investigated the use of ultrasound to suppress salmonid lice infections [131]. They compared the effects of sound frequencies of 9.3, 21, and 54 kHz on the parasite, and the experimental results showed that only exposure to 9.3 kHz significantly reduced lice infections. However, 9.3 kHz produced a maximum sound level of (220.6 dB), which could have an impact on surrounding organisms. To investigate whether ultrasonic waves cause secondary trauma in diseased fish during detection, Pottier et al. studied the detection of fish wounds using ultrasonic waves [132]. The experiments showed that the waveform had no effect on the percentage of fish injuries and could be a suitable method for the non-destructive detection of fish injuries.

Ultrasound is effective for the detection of pathogens, but it has limitations for the detection of fish diseases. Ultrasound cannot sense bleeding, blood, or body fluids in fish, nor can it sense them in fish with relatively small body cavities. Ultrasound may have an impact on surrounding fish when performing fish disease detection on individual culture ponds. In the future, an ultrasonic instrument could be developed to detect and inhibit the growth of parasites as a whole. After the detection of ultrasonic images, the wavelength could be automatically adjusted according to the types of parasite to inhibit the parasites and prevent the rapid spread of fish diseases.

### 3.6. Pathogen Diagnosis Based on Fluorescence Images

Fluorescence imaging has been used as an effective non-destructive testing technique for aquaculture fish disease detection. It has the advantages of low cost and the ability to image living organisms and organs. In detecting fish diseases, fluorescence imaging technology can identify parasites based on the different luminescence of fish tissues and parasites. Imaging based on principal component analysis can detect parasites, as indicated by Yang et al., who developed an ultraviolet fluorescence imaging technology based on principal component analysis and gray-value analysis to detect nematodes in fish products [133]. Using the blue–white light emitted by heterocyst larvae under UV light, the fish and larvae could be distinguished, and the overall detection rate of the experiment was 83.89%, which was an excellent detection result.

Fluorescence imaging technology can provide relevant information to determine the distribution and localization of bacteria in fish before bacterial infection. Ramachandran et al. developed non-invasive in vivo imaging of fish veins [134], which used a fluorescent probe to image the distribution of bacteria in fish. The limitation of this study is that the distributions of *Escherichia coli*, *Edwardsiella retardants*, *Vibrio alginolyticus*, and *Vibrio harveyi* were measured separately, and the internal organs of the fish showed stronger fluorescence signals in the in vitro images. To overcome the deviation of fluorescence imaging caused by pathogens and the internal organs of infected fish, and to make the fish appear transparent during detection, Ohnuma et al. investigated three tissue pigment removal techniques [135]. The experimental results showed that CUBIC processed the pigment more thoroughly than Clear and SeeDB. In the experiment, the researchers also determined that CUBIC 3D fluorescence imaging allowed for the 3D visualization of infected fish. The process of bacterial infection of fish can be detected using 3D fluorescence imaging to understand the mechanism of fish disease production. Kataoka et al. visualized the infection process of Edwardsiella in three dimensions using green fluorescent protein [136]. In this study, detailed and complete three-dimensional fish body information was obtained using holographic microscopy and film light microscopy. The experimental results revealed that the pathogens might first adhere to the skin of the fish, and then, enter various organs in the body through the gills and intestines.

Fish viruses can be detected by using the biological technique of PCR, which not only has ability to amplify DNA fragments in a short period of time, but also has strong amplification capacity, sensitivity, and specificity [137]. Considering the practicality of a diagnostic method that does not require sacrificing animals, fluorescence can be combined with PCR to achieve rapid detection of the number of viruses. Lopez-Jimena et al. proposed RT-PCR (Q-RT-PCR) to detect and determine the amount of viral hemorrhagic fever septicemia virus (VHSV) in rainbow trout organs [138], and this method proved that viral detection via RT-PCR could be performed on RNA extracted from blood samples. RT-PCR offers the possibility to automate virus detection and to detect viruses in asymptomatic fish. AA developed a nondestructive method for fish virus detection based on nested RT-PCR and speckle hybridization. Using the combination of nested RT-PCR and speckle hybridization, the detection rate of viruses in red ribbon fish in the experiment was up to 100%, and the detection rate of viruses in common fish was up to 94.4%.

Fluorescence imaging has higher sensitivity and real-time imaging capability for pathogen detection, but fluorescence imaging can be limited by the depth of detection, making it difficult to quantify precisely in vivo, and the detection of in vivo pathogens can be affected by the luminescence of internal organs. In the future, different degrees of luminescence of internal tissues and parasites could be enhanced, and fluorescence imaging could be used as an aid in image diagnosis to detect the scale of parasites in fish and to facilitate the application of appropriate drugs for the treatment of fish diseases.

### 3.7. Indirect Diagnosis Based on Electrochemical Sensors

Long-term studies on fish diseases have shown that the factors influencing fish diseases are numerous and complex, and various parameters can reflect the symptoms of fish diseases. Changes in water quality can lead to the growth of parasites which can be parasitic on the surface of the fish body or inside the fish body, causing pressure in the fish. To effectively treat fish diseases, it is necessary to detect parasites. Therefore, in addition to using image-processing technology, researchers have used various sensors based on different parameters to identify and evaluate fish diseases. Electrochemical genetic sensors provide a practical platform for the detection of eukaryotes as an analytical device that converts the detected signal into measurable parameters. Kuan et al. developed a highly sensitive and specific electrochemical genetic sensor [139]. This study used a combination of nano-latex particles and premixed hybridization analysis to rapidly diagnose epidemic ulcerative syndrome (EUS) in fish. The experiment showed that the genetic sensor was well adapted for EUS monitoring and diagnosis in aquaculture.

Electrochemical genetic sensors allow for rapid, inexpensive, and sensitive detection of fish diseases. In most studies, fish diseases are detected primarily through electrochemical DNA sensors [140]. In contrast, water-quality sensors can detect changes in the water environment by detecting dissolved oxygen, pH, water temperature, and other changes in the water that may be affected by fish health aspects, but not specific diseases. In the future, multiple intelligent sensors could be developed to detect not only changes in water quality but also parasites, fungi, and bacteria in the water. The information is also fused with the collected data and image diagnosis to improve the accuracy of fish disease diagnosis.

## 4. Conclusions and Future Perspectives

This article outlines the application of image-processing technology in aquaculture, especially the application of image-based fish disease diagnosis. The most important technical approaches to expert systems and automatic image-based diagnosis are summarized, and the direction of development of each diagnostic technique is discussed. Expert systems can provide highly accurate diagnoses for specific aquatic users with the help of information or images, while expert systems are limited in their speed of diagnosis and detection of unknown diseases by the rules of use and expert knowledge. Image processing combined with computer vision can provide a real-time, non-invasive, and economical technique for disease diagnosis, overcoming the limitations of expert systems to a certain extent. Camera images can detect disease on the surface of the fish, and microscopic images can provide details of minor changes in tissues within the fish to diagnose pathogens. However, both camera images and microscopic images can be limited by surface reflections and low-quality images. Hyperspectral images and ultrasound are not affected by the intensity of visible light, allowing the detection of fish pathogens in poorly lit and turbid water. However, their application accuracy is still generally low and relatively expensive.

The diversity and heterogeneity of fish diseases increased the difficulty in not only the diagnosis, but also the image acquisition of diagnostic methods in complex water environments. According to the actual situation at this stage, the focus of future research on fish disease diagnosis is as follows:(1)Image quality should be acquired and improved, and 3D models of fish diseases constructed. The 3D images of fish are captured using ultra-high-definition stereo cameras for all-around detection of diseased fish. In aquaculture, it is also possible to detect the location of fish bodies based on 3D images and to solve the problem of the heavy coverage of fish diseases in images. AR models are constructed using 3D images to allow fish disease experts to participate in the diagnosis to improve the accuracy of the diagnosis.(2)Standard and shared fish disease datasets should be established and existing automatic feature-extraction methods, such as convolutional neural networks (CNN), should be improved. Few expert systems are now utilized on the Internet, and there is a lack of uniform standards. The establishment of standard and shared fish disease datasets is indispensable. In the future, when diagnosing a large number of different diseases, it is crucial to unify the disease criteria in the dataset and combine the human–machine interface of the Internet to achieve social sharing of the dataset. Deep learning is used to analyze features that may be unique to various diseases to provide a viable and accurate method for diagnosis.(3)Using data fusion, data layer information fusion, feature layer information fusion, and decision layer information fusion could be used in different situations. Combining the information obtained from multi-parameter sensors, the accuracy of diagnosis is improved by the simultaneous detection of body surface and behavior, as well as internal tissues.

In summary, data fusion, 3D models, standard and shared datasets, and deep learning will be applied to fish disease diagnosis research, and standardization, automation, and intelligence are the future directions of fish disease diagnosis in aquaculture.

## Figures and Tables

**Figure 1 animals-12-02938-f001:**
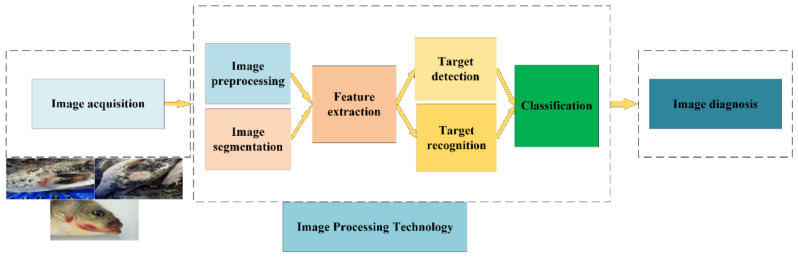
Illustration of the flow from acquisition of fish disease images through image-processing techniques to diagnosis.

**Figure 2 animals-12-02938-f002:**
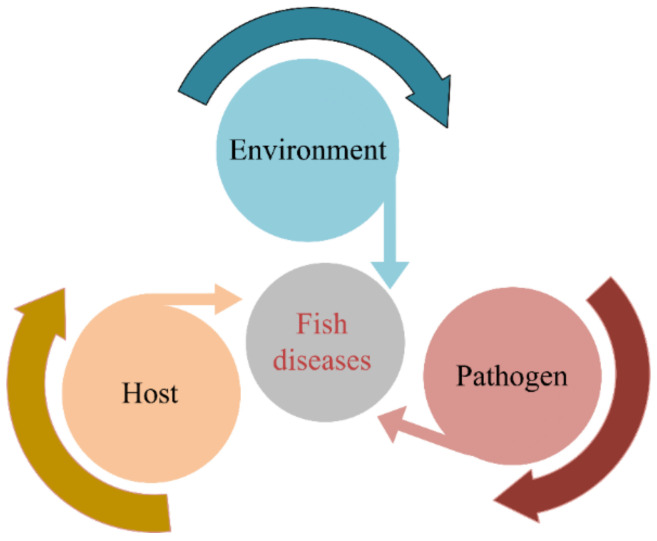
The interaction between host, pathogen, and environment.

**Figure 3 animals-12-02938-f003:**
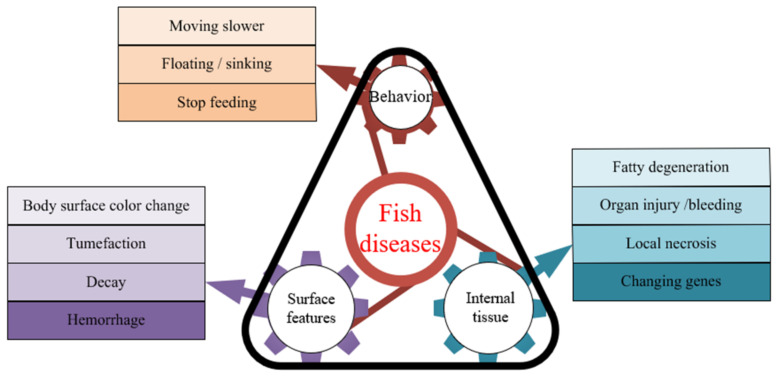
The surface and internal tissues as well as behavioral changes that may result from fish disease.

**Figure 4 animals-12-02938-f004:**
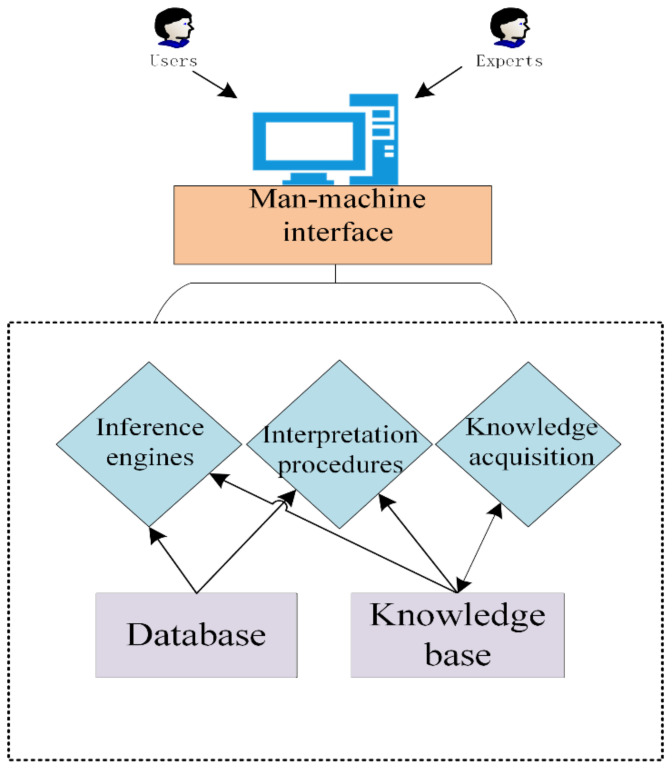
Illustration of the structure of the expert system and the flow of diagnosis.

**Figure 5 animals-12-02938-f005:**
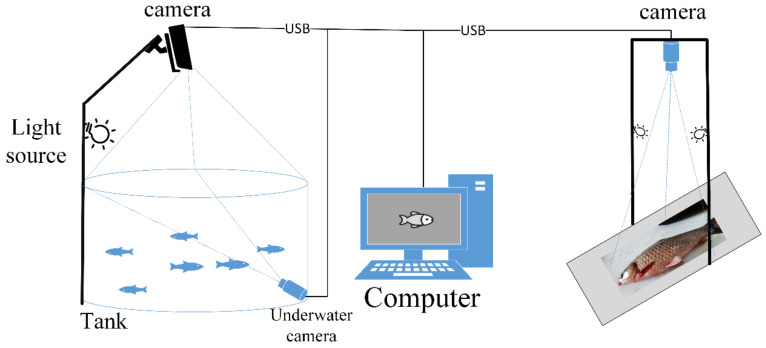
An architecture diagram of a computer vision-based fish disease diagnosis system.

**Table 1 animals-12-02938-t001:** The application of the clustering method, threshold method, and contour segmentation on aquatic environment.

Methods	Segmentation	Findings	Advantages	Disadvantages	References
Clustering method	Improved K-means	High accuracy and stability	Simple; fast	Unable to handle irregular data such as non-spherical; sensitive to the setting of initial value of k; sensitive to outliers	[41]
	Gabor filter K-means clustering	High-quality segmentation			[42]
	K-means clustering K-nearest neighbors	97.93%			[43]
	K-means algorithm	Shortens algorithm running time			[44]
Threshold method	Change threshold to refine segmentation	90.86%	Simple; not affected by image contrast and brightness changes under certain conditions	It is difficult to use a real-time system when there is noise, and a large number of calculations need an optimal threshold	[45]
	Three-dimensional entropy	Algorithm efficiency is greatly improved			[46]
	Cumulative threshold	Accurate segmentation of gills			[47]
Based on contour	Active contour segmentation	Improved efficiency and accuracy	Overcomes the shortcomings of other methods of segmenting images in small and continuous space, and has better regional characteristics	It is easy to cause excessive segmentation of the image	[48]
	Segmentation of contour	>90%			[49]

**Table 2 animals-12-02938-t002:** The application of feature extraction in aquaculture fish identification and behavior detection.

Fields	Features	Methods	Accuracy	Advantage/Disadvantage	References
Identification	Color Shape	Feedforward neural network	>96%	Multi-feature selection based on color, shape, texture, etc. improves accuracy, but the time required for feature extraction is longer	[50]
	Color Shape Texture	Deep network model	98.64%		[51]
	Shape Texture Color Head shape	Deep convolutional neural network	>90%		[52]
Motion detection	Static motion Space–time movement	Space–time local kinematic model (STLKP)	Automatic classification of abnormal and normal motion	Motion feature can detect motion state, single-feature detection accuracy is low	[53]
		Local Kinematic Shape Pattern (LKSP)			[54]

**Table 3 animals-12-02938-t003:** The accuracy of the different classifiers in detection.

Application	Methods	Accuracy	References
Carp classification	SVM	94%	[69]
	Naive Bayes	96.80%	
Classification of breeding and non-breeding fish	CNN deep learning framework	89%	[70]
	SVM	84.78%	
	Naive Bayes	87%	
	MLP	83.7%	
	Random forest	86.95%	
	Decision tree	81.52%	
Rainbow trout classification	Convolutional neural network	96.51%	[71]
Underwater fish classification	Backpropagation neural network	90.24%	[72]

**Table 4 animals-12-02938-t004:** The application of image-processing technology in researching diseases in zebrafish.

Application	Methods	Diseases	References
Neural system detection	Image analysis	Alzheimer’s disease (AD)	[112]
	Computer vision and biological image analysis	Evaluation of neuronal cells	[113]
	Image analysis	Compounds formed by oligodendrocyte	[114]
	Image system and histology	Nerve chronic infection	[115]
Embryo	Improved image segmentation	Embryo abnormality	[116]
	Automatic image segmentation	Yolk absorption non-performing	[117]
Behavior	Machine learning	Parkinson’s disease	[118]

## Data Availability

Not applicable.

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
