# Peer review of "Advanced Techniques for the Intelligent Diagnosis of Fish Diseases: A Review"

_animals, 2022, doi:10.3390/ani12212938_

Round 1

Reviewer 1 Report

In this article, authors have presented a review on application of image processing technology in aquaculture especially the application of image-based fish disease diagnosis. Various techniques in the literature have been presented. Writing is good. Future direction for research are also presented. This article is acceptable for publication.

Author Response

Response to Reviewers

Dear editors and reviewers:

Thank you for your letter and the reviewers’ comments on our manuscript entitled " Advanced techniques for the intelligent diagnosis of fish diseases: a review " (animals-1963404). Those comments are very helpful for revising and improving our paper, as well as the important guiding significance to other research. We have studied the comments carefully and made corrections which we hope meet with approval. As for the language of the manuscript, we have arranged for a domain expert from an English-speaking country to help proofread and improve the manuscript, and the manuscript has been substantially revised in terms of language. The main corrections are in the manuscript.Please see the "author-coverletter-22969766.v1.docx" word document for detailed response.

Reviewer 2 Report

In this article, the authors detailly reviewed image processing technology and image-based fish disease detection methods used in diagnosis of fish diseases, analyzed the benefits and drawbacks of each diagnostic approach in different environments, and finally discussed and summarized the potential techniques for the applications of fish disease identification and diagnosis. This study is interesting and worthy of publication in Animals after some issues are resolved.

1. Figure 2 shows the interaction between host, pathogen, environment and fish diseases. But in the caption, “parasite” but not “pathogen” was used.

2. Figure 3 shows the surface and internal tissue as well as behavioral changes that may result from the fish disease. The author described “The detection of pathogens is mainly based on the internal tissues of the fish”, they just focus on the changes of internal organization (such as degeneration, necrosis…). However, “the detection of pathogens” should pay more attention on the pathogen identification.

3. line 757-758, “Wortberg et al used …… to classify the [142]”, please complete this sentence “to classify the ?”.  

Moreover, in this paragraph (line 756-774), I think the authors want to describe the detection of bacteria, however, the detection of parasites, allergens, and fine albumin were also included(line 771-774), it’s a bit of a mess.

The title of “3.4. Pathogen detection based on spectral images”, other pathogens except the mentioned parasites and bacteria might be introduced, such as “virus detection”.

4. the Latin ‘scientific’ names of fish and bacteria should be in italics.

 line 646: Oncorhynchus nerka; line 790: Carassius auratus; line[1]827: Escherichia coli, Edwardsiella retardants, Vibrio alginolyticus, and Vibrio harveyi.

5. In 3.6. Parasite diagnosis based on fluorescent images. The title is about Parasite diagnosis”, but the bacteria were described in line 826-827.

6. In 3.7. Indirect diagnosis based on electrochemical sensors”,however, direct detection of the VHSVwere described.

Author Response

Response to Reviewers

Dear editors and reviewers:

Thank you for your letter and the reviewers’ comments on our manuscript entitled " Advanced techniques for the intelligent diagnosis of fish diseases: a review " (animals-1963404). Those comments are very helpful for revising and improving our paper, as well as the important guiding significance to other research. We have studied the comments carefully and made corrections which we hope meet with approval. As for the language of the manuscript, we have arranged for a domain expert from an English-speaking country to help proofread and improve the manuscript, and the manuscript has been substantially revised in terms of language. The main corrections are in the manuscript and the responses to the reviewers’ comments are as follows.Please see the word document for detailed response.

Replies to the reviewers’ comments:

-Reviewer

 - In this article, the authors detailly reviewed image processing technology and image-based fish disease detection methods used in diagnosis of fish diseases, analyzed the benefits and drawbacks of each diagnostic approach in different environments, and finally discussed and summarized the potential techniques for the applications of fish disease identification and diagnosis. This study is interesting and worthy of publication in Animals after some issues are resolved.

 [1]. Figure 2 shows the interaction between host, pathogen, environment and fish diseases. But in the caption, “parasite” but not “pathogen” was used.

Response: Thank you for your valuable comments.

According to your suggestions, we have revised the caption of Figure 2. We have also checked the information to see the meaning of pathogen and parasite in more detail. Pathogen: a collective term for microorganisms and parasites that cause disease; parasite: generally a worm that lives in animals or humans. See line 125 in the manuscript for details.

Figure 2. The interaction between host, pathogen, and environment.

[2]. Figure 3 shows the surface and internal tissue as well as behavioral changes that may result from the fish disease. The author described “The detection of pathogens is mainly based on the internal tissues of the fish”, they just focus on the changes of internal organization (such as degeneration, necrosis…). However, “the detection of pathogens” should pay more attention on the pathogen identification.

Response: Thank you for your valuable comments.

According to your suggestions, we have revised the description of Figure 3 in the manuscript: Images can be used in aquaculture to diagnose fish diseases by detecting changes in hosts, pathogens, and the environment. Most of the existing studies have shown that the detection of hosts is mainly based on changes occurring from the surface of the fish and the detection of pathogens is mainly based on changes in the internal tissues of the fish, such as deformation, necrosis, decay, and bleeding. But the methods of detecting pathogens could only roughly detect the existence of pathogens. There are many different types of pathogens in fish and the identification of pathogens is usually dependent on biochemical pathogen detection methods. See lines 378 and 117 of the manuscript for details.

[3]. line 757-758, “Wortberg et al used …… to classify the [142]”, please complete this sentence “to classify the ?”. 

Response: Thank you for your valuable comments.

According to your suggestions, we have complemented the research by Wortberg et al on lines 757-758: Wortberg et al used biochemical methods with Fourier transform infrared spectroscopy (FT-IR) combined with artificial neural network analysis to characterize the strains, while DNA sequencing was performed on some strains using biochemical methods. The results finally confirmed that the differentiation method with the FT-IR module was 97.4% correct. See line 119 in the manuscript for details. See lines 701 and 705 of the manuscript for details.

- Moreover, in this paragraph (line 756-774), I think the authors want to describe the detection of bacteria, however, the detection of parasites, allergens, and fine albumin were also included(line 771-774), it’s a bit of a mess.

Response: Thank you for your valuable comments.

According to your suggestion, we have rechecked "3.4. Pathogen detection based on spectral images" and realized that allergen and fine albumin detection is not suitable for this subsection, so we decided to remove allergen and fine albumin detection. See lines 700 and 716 of the manuscript for details.

- The title of “3.4. Pathogen detection based on spectral images”, other pathogens except the mentioned parasites and bacteria might be introduced, such as “virus detection”.

Response: Thank you for your valuable comments.

According to your suggestion, we have added to "3.4. Pathogen detection based on spectral images" a subsection in which we have included virus detection.

Viruses as a non-cellular organism, and each virus contains only one type of nucleus, either DNA or RNA. Spectral images have been confirmed for the detection of viruses, but accurate detection of fish viruses is now mainly by PCR. Virus detection based on spectral images has been applied to botany, and good progress has been made. For example, Morellos et al used spectroscopy to obtain spectral images of healthy and diseased tomato leaves, and then used the XY-fusion network (XY-F) and the multilayer perceptron with automated relevance determination (MLP-ARD) to estimate disease presence and virus load in tomato leaves, respectively. The results showed that the MLP-ARD classified machine outperformed the XY-F network overall, with an overall accuracy of 92.1%, compared to 88.3% for XY-F. In the future, the quality of the spectral images will be improved by optimizing the fish habitat, lowering the impact of the complex environment in the spectral images, and improving the quality of the spectral images through suitable image processing technology. In this way, better spectral images can be obtained and used for the detection of fish viruses. See lines 717 and 730 of the manuscript for details.

[4]. the Latin ‘scientific’ names of fish and bacteria should be in italics.

- line 646: Oncorhynchus nerka; line 790: Carassius auratus; line[1]827: Escherichia coli, Edwardsiella retardants, Vibrio alginolyticus, and Vibrio harveyi.

Response: Thank you for your valuable comments.

Your suggestions have been very helpful to us. We have revised the Latin "scientific" names of fish and bacteria in lines 646, 790, and 827, and italicized them all.

Balaban et al. developed an image analysis method to quantify the Quantification of Gaping, Bruising, and Blood Spots of sockeye salmon (Oncorhynchus nerka) fillets. See line 592 in the manuscript for details.

Navot et al. studied the effect of ultrasound on the immunization of goldfish (Carassius auratus). See line 746 in the manuscript for details.

The limitation of this study is that the distribution of Escherichia coli, Edwardsiella retardants, Vibrio alginolyticus, and Vibrio harveyi was measured separately, and the internal organs of the fish showed stronger fluorescent signals in the in vitro images. See line 783 in the manuscript for details.

[5]. In 3.6. Parasite diagnosis based on fluorescent images. The title is about “Parasite diagnosis”, but the bacteria were described in line 826-827.

Response: Thank you for your valuable comments.

According to your suggestion, we have revised “3.6. Parasite diagnosis based on fluorescent images” with one paragraph for parasite detection and the other for bacterial detection. And we have also added a paragraph on virus detection based on fluorescent images, so we have revised the title of 3.6 to “Pathogen diagnosis based on fluorescent images”. Detection of viruses is the third subparagraph:

Fish viruses can be detected by using the biological technique of PCR, which not only has the ability to amplify DNA fragments in a short period of time, but it also has strong amplification capacity, sensitivity and specificity. Considering the practicality of a diagnostic method that does not require sacrificing animals, fluorescence can be combined with PCR to achieve a rapid detection of the number of viruses. Lopez-Jimena et al proposed an RT-PCR (Q-RT-PCR) to detect and determine the amount of viral hemorrhagic fever septicemia virus (VHSV) in rainbow trout organs, and this method proved that viral detection by RTPCR could be performed on RNA extracted from blood samples. RT+PCR offers the possibility to automate virus detection and to detect viruses in asymptomatic fish. AA developed a nondestructive method for fish virus detection based on nested RTPCR and speckle hybridization. Using the combination of nested RT-PCR and speckle hybridization, the detection rate of virus in red ribbon fish in the experiment was up to 100%, and the detection rate of virus in common fish was up to 94.4%. See lines 798 and 810 of the manuscript for details.

Finally, we would like to thank you again.

 [6]. In “3.7. Indirect diagnosis based on electrochemical sensors”, however, “direct detection of the VHSV” were described.

Response: Thank you for your valuable comments.

Your suggestions have been very helpful to us. We have rechecked "3.7. Indirect diagnosis based on electrochemical sensors" and agree with your suggestion that “direct detection of the VHSV” is not appropriate here as it may cause misunderstanding to the reader. So we have decided to remove the “direct detection of the VHSV”.

Reviewer 3 Report

Dear Authors,

the Manuscript ID: animals-1963404 is original and could be of interest for Animals’ readers some structural changes are requested.

Infectious diseases represent one of the main issues for aquaculture. Traditional methods for fish diseases detection are usually related to the knowledge of fish owner and fish health experts and laboratory analysis. This could imply lack of accuracy or time-consuming procedures.

Considering the high economic impact of diseases due to direct and indirect costs, the high capacity of spreading of the pathogens through waters, rapid and more accurate methods are requested. Image-based methodologies could fill this diagnostic gap. Moreover, methods with a prognostic value applicable to alive fish could be desirable than methods applicable only on dead fish or causing fish injuries during the sampling.

I have concern mainly related to the structure of this review:

considering the target readers of Animals, the paper should be more focussed on intelligent diagnosis method of fish diseases as reported in paragraph 3 more than in the detailed technical description of image processing technology as reported in paragraph 2.

Methods should be reported grouping conservative methods applicable on alive fish or methods applicable only on dead fish or causing injuries to the fish (diagnosis on internal tissues of the fish using microscopy).

Minor concerns are related to the need of implementation of some paragraphs:

Introduction

Line 47: not only parasites are related to zoonosis, but also bacteria such as Mycobacterium, S. iniae, Aeromonas; an implementation of the sentence and the related references is requested.

Lines 68-75: advantages and disadvantages of methods are reported but costs are not cited.

Paragraph 2

Description of criticalities related to image analysis should be better explained to better justify all the steps of the image technology described in the sub-paragraphs.

2.1

Please explain better in which condition should be taken the fish for image acquisition: is the fish in aquatic natural environment or outside and under sedation?

2.3.2

Line 260: Please change water with environment

 2.4

Line 279: table 2 is not only related to fish detection as reported here but to fish freshness, identification and behaviour detection.

Paragraph 3

Figure 3: Please change the surface characteristics reported; body surface colourless in incorrect because hypermelanosis is a very common symptoms of infected fish. The sentence could be change as alteration of body surface colour

Moreover, considering that freshness is not related to fish disease diagnosis I suggest removing these data.

Different typo errors are present in the text (i.e. lines 140, 238, 490) and the use of English language should be revised.

Author Response

Response to Reviewers

Dear editors and reviewers:

Thank you for your letter and the reviewers’ comments on our manuscript entitled " Advanced techniques for the intelligent diagnosis of fish diseases: a review " (animals-1963404). Those comments are very helpful for revising and improving our paper, as well as the important guiding significance to other research. We have studied the comments carefully and made corrections which we hope meet with approval. As for the language of the manuscript, we have arranged for a domain expert from an English-speaking country to help proofread and improve the manuscript, and the manuscript has been substantially revised in terms of language. The main corrections are in the manuscript and the responses to the reviewers’ comments are as follows.Please see the "author-coverletter-23047061.v1.docx" word document for detailed response.

Replies to the reviewers’ comments:

-Reviewer

 - the Manuscript ID: animals-1963404 is original and could be of interest for Animals’ readers some structural changes are requested.

Infectious diseases represent one of the main issues for aquaculture. Traditional methods for fish diseases detection are usually related to the knowledge of fish owner and fish health experts and laboratory analysis. This could imply lack of accuracy or time-consuming procedures.

Considering the high economic impact of diseases due to direct and indirect costs, the high capacity of spreading of the pathogens through waters, rapid and more accurate methods are requested. Image-based methodologies could fill this diagnostic gap. Moreover, methods with a prognostic value applicable to alive fish could be desirable than methods applicable only on dead fish or causing fish injuries during the sampling.

I have concern mainly related to the structure of this review:

considering the target readers of Animals, the paper should be more focussed on intelligent diagnosis method of fish diseases as reported in paragraph 3 more than in the detailed technical description of image processing technology as reported in paragraph 2.

Response: Thank you for your valuable comments.

Your suggestion is very helpful and according to your suggestion, the manuscript should focus more on intelligent diagnosis method of fish diseases in paragraph 3, considering the target audience of animal journals, and we have revised paragraph 2. We have summarized the individual subsections in paragraph 2 in more detail and have reduced the proportion of paragraph 2 in the manuscript so that the reader will be able to quickly understand the process and methods of image processing technology. In addition, we have removed some sections of image processing technology that are not related to fish disease detection, such as the detection of freshness. The pre-revised paragraph 2 is 86 lines to 449 lines, while the revised paragraph 2 is 86 lines to 372 lines.

Methods should be reported grouping conservative methods applicable on alive fish or methods applicable only on dead fish or causing injuries to the fish (diagnosis on internal tissues of the fish using microscopy).

Response: Thank you for your valuable comments.

According to your suggestion, we have added a classification of disease detection methods for live and dead fish in lines 394 and 413:

These methods can be divided into two groups according to whether the detection method is damaging to the fish. The damaging ones include expert systems, microscopic images, fluorescent images, and ultrasonic images; the nondamaging ones camera images, spectral images, and indirect detection by sensors. And according to the detection method of live or dead fish will be divided into two groups, the detection of live fish includes the detection of camera images, spectral images, fluorescent images, ultrasonic images, and indirect detection by sensors; the detection of dead fish includes expert systems, microscope images.

This is because camera images and spectral images can be used to detect fish diseases using images taken in the water, so they are mainly used for live fish disease detection, and they do not cause damage to the fish’s body. The expert system needs to detect fish diseases through eight parts: body surface, head, gills, abdomen, scales, fins, muscles, internal organs, etc. Therefore, the expert system is mainly applied to dead fish disease detection, and it has damage to the fish body. Microscope images need to dissect the fish body to obtain internal tissue images to detect diseases, so microscope images are mainly applied to dead fish and have damage to the fish body. Fluorescence images are used to detect live fish because the fluorescent agent in the water flows into the tissue of the fish, but the fluorescent agent inside the fish can cause some damage to the fish. The detection of ultrasonic images is mainly applied to the detection of disease in live fish underwater in a complex environment, but the wavelength of ultrasonic waves can cause damage to the fish’s body.

Minor concerns are related to the need of implementation of some paragraphs:

Introduction

Line 47: not only parasites are related to zoonosis, but also bacteria such as Mycobacterium, S. iniae, Aeromonas; an implementation of the sentence and the related references is requested.

Response: Thank you for your valuable comments.

According to your suggestion, we have added the effect of bacteria on fish and people's health in 46: Worse yet, it is not only parasites that can cause harmful effects on human health by diseased fish [6], but fish with bacteria that can also infect humans with diseases such as Salmonella [7]. See lines 46 and 48 of the manuscript for details.

Lines 68-75: advantages and disadvantages of methods are reported but costs are not cited.

Response: Thank you for your valuable comments.

According to your suggestion, we have revised the advantages and disadvantages of lines 68-75 and added the advantages of low cost, simple operation, high sustainability, and no pollution to water bodies.

The use of camera images, microscopic images, spectral images, ultrasound images, and fluorescent images has been shown to provide feasibility for fish disease diagnosis, and the combination of image processing technology and computer vision can provide non-destructive, automatic, rapid, real-time diagnosis of fish diseases, and low cost, simple operation, high sustainability, and no pollution to water bodies. See lines 68 and 73 of the manuscript for details.

Paragraph 2

Description of criticalities related to image analysis should be better explained to better justify all the steps of the image technology described in the sub-paragraphs.

2.1

Please explain better in which condition should be taken the fish for image acquisition: is the fish in aquatic natural environment or outside and under sedation?

Response: Thank you for your valuable comments.

Your suggestion is very helpful, according to your suggestion, we have added the classification of image acquisition in lines 108 & 115: According to different acquisition methods fish disease image acquisition could be divided into two categories, one is in the aquatic natural environment and the other is under external human intervention. Image acquisition in the aquatic natural environment is not affected by human interference, and these images are mainly used to detect the disease of fish by abnormal fish surface or abnormal fish swimming, which mainly include: camera images, spectral images, and ultrasonic images. Image acquisition under external human intervention can cause damage to the fish, and this category mainly includes: expert systems, microscope images, and fluorescence images.

This is because expert systems and microscope images require the detection of fish internal tissues, and fluorescence images are acquired by pouring fluorescent agents into the water in experimental environments and then collecting fluorescence images of fish internal tissues when the fish is sedated. See lines 108 and 115 of the manuscript for details.

2.3.2

Line 260: Please change water with environment

Response: Thank you for your valuable comments.

According to your suggestion, we have changed water with environment in line 260. See line 245 of the manuscript for details.

 2.4

Line 279: table 2 is not only related to fish detection as reported here but to fish freshness, identification and behaviour detection.

Response: Thank you for your valuable comments.

According to your suggestion, we also realized that the fish freshness detection really cannot be put in the fish disease diagnosis, so we have accepted your opinion and removed the fish freshness detection in Table 2. See line 269 of the manuscript for details.

Paragraph 3

Figure 3: Please change the surface characteristics reported; body surface colourless in incorrect because hypermelanosis is a very common symptoms of infected fish. The sentence could be change as alteration of body surface colour。

Response: Thank you for your valuable comments.

According to your suggestion, we have revised Figure 3, as shown in the figure below.

Figure 3. Shows the surface and internal tissue as well as behavioral changes that may result from the fish disease.

Moreover, considering that freshness is not related to fish disease diagnosis I suggest removing these data.

Response: Thank you for your valuable comments.

According to your suggestion, we also realized that the fish freshness detection really cannot be put in the fish disease diagnosis, so we have accepted your opinion and removed the fish freshness detection in Table 2. See line 269 of the manuscript for details.

Different typo errors are present in the text (i.e. lines 140, 238, 490) and the use of English language should be revised.

Response: Thank you for your valuable comments.

According to your suggestion, we have rechecked the manuscript and corrected the typos, and we have arranged for a domain expert from an English-speaking country to help proofread and improve the manuscript, and the manuscript has been substantially revised in terms of language.

Round 2

Reviewer 3 Report

Dear Authors,

thank you for revising the review in accordance with the suggestions.